# Atomistic-Level Structural Insight into Vespa Venom (Ves a 1) and Lipid Membrane Through the View of Molecular Dynamics Simulation

**DOI:** 10.3390/toxins17080387

**Published:** 2025-07-31

**Authors:** Nawanwat Chainuwong Pattaranggoon, Withan Teajaroen, Sakda Daduang, Supot Hannongbua, Thanyada Rungrotmongkol, Varomyalin Tipmanee

**Affiliations:** 1Programme in Bioinformatics and Computational Biology, Graduate School, Chulalongkorn University, Bangkok 10330, Thailand; nawanwat.p@rsu.ac.th; 2Faculty of Medical Technology, Rangsit University, Muang Pathumthani, Pathumthani 12000, Thailand; 3Health Service Center for Medical Technology and Physical Therapy, Faculty of Associated Medical Sciences, Khon Kaen University, Khon Kaen 40002, Thailand; withan_t@kkumail.com; 4CharoenLuck Group, Khon Kaen 40002, Thailand; 5Division of Pharmacognosy and Toxicology, Faculty of Pharmaceutical Sciences, Khon Kaen University, Khon Kaen 40002, Thailand; sakdad@kku.ac.th; 6Center of Excellence in Computational Chemistry (CECC), Department of Chemistry, Faculty of Science, Chulalongkorn University, Bangkok 10330, Thailand; supot.h@chula.ac.th; 7Center of Excellence in Structural and Computational Biology, Department of Biochemistry, Faculty of Science, Chulalongkorn University, Bangkok 10330, Thailand; 8Department of Biomedical Sciences and Biomedical Engineering, Faculty of Medicine, Prince of Songkla University, Hat Yai, Songkhla 90110, Thailand

**Keywords:** molecular dynamics simulation, Ves a 1, *Vespa affinis*, lipid membrane, voxilaprevir, protein-membrane interaction, enzyme inhibition

## Abstract

This study used all-atom molecular dynamics simulations to investigate the structural dynamics of Ves a 1, a phospholipase from *Vespa affinis* venom, and its interactions within a lipid membrane environment, both alone and in the presence of the inhibitor voxilaprevir. Simulations conducted over 1 µs for triplicate runs demonstrated system stability and convergence of structural properties. Our findings reveal that Ves a 1 engages in dynamic interactions with the lipid bilayer, involving key regions such as its lids, catalytic triad, and auxiliary site. The presence of voxilaprevir was observed to subtly alter these membrane interaction patterns and influence the enzyme’s catalytic area, reflecting the inhibitor’s impact within its physiological context. These results emphasize the crucial role of the lipid bilayer in shaping enzyme function and highlight voxilaprevir as a promising candidate for further inhibitor development, offering vital insights for rational drug design targeting membrane-associated proteins.

## 1. Introduction

Insects represent the most diverse group of animals and include a significant proportion of venomous arthropods. Among them, members of the order *Hymenoptera*—specifically those belonging to the families *Apidae* (bees), *Formicidae* (ants), and *Vespidae* (wasps)—are the primary agents responsible for sting-induced allergic reactions in humans [1]. In Southeast Asia, the Thai banded tiger wasp (*Vespa affinis*) is recognized as one of the most dangerous vespid species, with envenomation resulting in severe clinical outcomes [2].

The venom of *V. affinis* contains several toxic proteins, among which phospholipase $A_{1}$ (Vespapase or Ves a 1) is considered particularly lethal [2]. Ves a 1 is known to induce not only allergic responses and anaphylaxis but also significant biological effects, including membrane disruption [2,3], haemolysis [4], inflammation [5], and platelet aggregation that can lead to thrombosis [6]. Notably, *V. affinis* is responsible for the highest number of fatal incidents due to wasp stings in the region [7,8]. Reports describe severe systemic toxicity, including multiorgan failure and death, in patients stung by large numbers of wasps [7,8]. Additional cases document localized symptoms such as inflammation, hemorrhage, and necrosis following dermal exposure to venom [9]. While supportive therapies such as epinephrine and corticosteroids are commonly administered, they may not be suitable for hypertensive patients. In critical cases, advanced interventions, including respiratory and cardiovascular support, haemodialysis, and haemoperfusion, are required [10].

Due to the absence of a crystal structure for Ves a 1 in protein databases, the protein’s three-dimensional model was constructed using bioinformatics tools. This model was validated via comparison to the X-ray crystal structure of vPLA2 [11], which shares 93% sequence similarity [12,13]. Ves a 1 contains a conserved catalytic triad (Ser-X-Asn-X-His) essential for its enzymatic activity [3,14], as well as an auxiliary site believed to assist in membrane binding [15]. As a member of the ${\mathrm{PLA}}_{1}$ family, Ves a 1 targets the *sn-1* position of phospholipids within the lipid bilayer, catalyzing their hydrolysis and generating lysophospholipids and free fatty acids, which compromise membrane integrity [4,5,6,12]. The auxiliary site, while not catalytic, appears to play a critical role in anchoring the enzyme to the membrane surface [13,15].

The repurposed drug became of interest in drug discovery, including an enzyme inhibitor [16,17,18,19]. We also adopted this concept in the study of Ves a 1 inhibitor exploration. Our previous investigation of Ves a 1 in a solution with 2056 FDA-approved drugs and a ligand control (1,2-Dimyristoyl-sn-glycero-3-phosphocholine or DMPC) revealed that voxilaprevir was one of the top candidates for binding Ves a 1 at both the catalytic and auxiliary sites [13]. We decided to use voxilaprevir, a protease inhibitor, as a repurposed drug candidate. Despite these findings, the detailed interaction and behaviors of Ves a 1 under lipid membrane bilayer remain insufficiently understood. Membrane environment inclusion is a key factor for fully understanding membrane-embedded proteins, both in ligand-free and ligand-bound states [20,21,22].

To address this knowledge gap, we employed all-atom molecular dynamics (MDs) simulations to investigate the structural information of Ves a 1 in the presence of a lipid bilayer. Furthermore, we examined how the binding of voxilaprevir—a proposed protease inhibitor [13]—affects the conformational dynamics and membrane interaction of Ves a 1. The aim of this study is also to provide structural insights into how voxilaprevir binds to Ves a 1 and understand whether its binding may influence its action on the lipid substrate, particularly in membrane-associated environments, thereby supporting the potential development of Ves a 1-targeted inhibitors.

## 2. Computational Methods

### 2.1. Parameterization

The amino acid sequences of Ves a 1 were taken from the Uniprot database (UniProtKB code: P0DMB4) and then submitted to AlphaFold 2 [23,24] for structure prediction. The 3D structure of Ves a 1 was predicted based on the UniProt entry (ID: P0DMB4), with the corresponding AlphaFold model ID: AF-P0DMB4-F1-v4. The per-residue confidence score (pLDDT) indicated very high confidence (pLDDT > 90). The prediction result was provided in the Appendix A. Using the APBS web service [25], the protonation states of all ionisable amino acids were assigned at pH 7.4.

CHARMM-GUI Membrane Builder [26,27] was used to build protein–membrane systems. To address this, the bilayer membrane utilized in the simulation should be consistent with the organism. Numerous human membrane models were employed, and the component’s capacity to modify membrane fluidity may result in inconsistent results across various models. However, to avoid these issues, the simplified bilayer model is still sufficient for the molecular dynamics investigation of the biological system [20,22].

To represent this model, we implemented the simple bilayer membrane model (80% DMPC and 20% DOPG). The simple lipid bilayer model was built based on a mixture of 80% of phosphatidylcholines (DMPC) and 20% of phosphatidylglycerol (DOPG), similar to previous studies [28]. Both simple lipid bilayers were symmetric in terms of the lipid composition of the upper and lower leaflets. Ves a 1 was initially placed about 15 Å above the upper leaflet of the bilayers (Figure 1B), and then the system was solvated using a TIP3P water model [29]. NaCl was used as a counter-ion, added to neutralize the charge of the systems to a concentration of 0.15 M.

The structure of Ves a 1 + voxilaprevir was modeled by superimposing it on the ligand-free Ves a 1 placed above the membrane. The final MD snapshot of Ves a 1 + voxilaprevir from the previous study was selected for superimposition in order to produce a Ves a 1 + voxilaprevir + membrane complex, which served as the initial coordinate for the three MD runs. The system preparation process for both the ligand-free and ligand-bound Ves a 1 structures is illustrated in Figure 2.

### 2.2. All-Atom Molecular Dynamics (MDs) Simulation

The simulation was carried out using the PMEMD module implemented in the AMBER20 package [30]. After the system was minimized to remove unusual inter-atomic contacts, the system was equilibrated in a constant number (N), volume (V), and temperature (T) (NVT) ensemble. Temperature in each simulated system was gradually increased up to 310 K for 200 ps with an application of a harmonic positional restraint of 100 kcal/mol ${Å}^{2}$ to the $C_{\alpha }$ atoms of the protein. A cut-off distance for non-bonded interactions was set to 12 Å, while the particle mesh Ewald summation method was employed to treat the electrostatic interactions. The SHAKE algorithm was used to constrain all covalent bonds involving hydrogen atoms. A 2 fs simulation time step was used throughout the MD simulation.

The temperature and pressure were controlled by the Langevin thermostat, with a collision frequency of 1/ps, and the Berendsen barostat, with a pressure-relaxation time of 1 ps. Subsequently, each simulated system was performed under the periodic boundary condition with the isothermal–isobaric of the constant number (N), pressure (P), and temperature (T) (NPT) ensemble with a temperature of 310 K and a pressure of 1 atm until reaching 1 µs without any restraint, as per previous studies [31]. The total binding free energy (Δ$G_{Bind}^{SIE}$) [32,33] was calculated using the solvated interaction energy (SIE) method, which was calculated using the following Equation (Equation 1) [13]:(1)$$\Delta G_{Bind}^{SIE}=\alpha \left(\underset{Nonpolar}{\underset{︸}{\Delta E_{vdW}+\gamma \Delta SA}}+\underset{Electrostatic}{\underset{︸}{\Delta E_{Ele}+\Delta G_{RF}}}\right)+C$$
where $\Delta E_{vdW}$ is van der Waals, γΔSA is the cavity, $\Delta E_{Ele}$ is electrostatic, and $\Delta G_{RF}$ is the reaction field. $\Delta E_{vdW}$ and $\Delta E_{Ele}$ are denoted as intermolecular energies in the bound state. The coefficients used in every calculation are α, γ, and *C* as 0.105, 0.013, and −2.89, respectively, similar to the previous studies [34].

## 3. Results and Discussion

In this study, we focused on Ves a 1 under the bilayer membrane condition. The membrane-free Ves a 1 was investigated before in a previous study [13]. The computer simulations, which looked at individual atoms and ran for 1 µs, helped us understand how Ves a 1 changes its shape with and without the drug voxilaprevir within a membrane environment. Figure 3 presents the root-mean-square displacement (RMSD), serving as a crucial indicator of the structural stability and convergence of our molecular dynamics simulations. The black, gray, and light-gray lines correspond to three distinct simulation runs (Run#1, Run#2, and Run#3). The upper panels illustrate the RMSD for the entire system, while the lower panels specifically depict the RMSD of Ves a 1 alone and Ves a 1 in a complex with voxilaprevir. The vertical red dashed line in Figure 3 indicates the approximate point at 800 ns after which the systems reached a stable equilibrium.

**System stability:** In the simulation conditions involving the membrane (Ves a 1 + Membrane and Ves a 1 + Membrane + Voxilaprevir), the RMSD values rapidly increased during the initial phases (approximately 0.0–0.2 µs). This increase showed that the systems were making structural adjustments to find a stable arrangement. Specifically, the RMSD values for the entire system (Ves a 1 + Membrane; Ves a 1 + Membrane + Voxilaprevir) reached approximately 60–70 Å, while those for Ves a 1 alone or complexed with voxilaprevir typically ranged from 2–3 Å. After this initial period, the RMSD values for all runs generally settled down and showed relatively stable changes for the remaining 1.0 µs simulation time. This steady behavior, with small movements around an average value, means the systems reached a balanced state. The consistency across the triplicate runs (Run#1, Run#2, and Run#3) further confirms the robustness and reproducibility of the simulations.

The RMSD graphs became stable, especially after approximately 0.8 µs (the last 200 ns of the 1.0 µs simulation), indicated by the vertical red dotted line, which tells us that the systems had explored their possible shapes enough and reached a steady balance. Because of this, we carefully chose the last 200 ns of data from each simulation run for further detailed analysis. This choice ensures that all later analyses—such as measuring distance, checking the catalytic area, and estimating binding energies—are based on the stable shapes of Ves a 1 and its complex with voxilaprevir in the presence of the membrane. This approach helped us reduce the effects of initial system adjustments and made our results as accurate as possible.

Figure 4A shows heatmaps that illustrate the shortest distances between specific amino acids of Ves a 1 and the phosphate groups of the lipid membrane over time. The left graph shows how Ves a 1 by itself interacts with the membrane, while the right graph shows Ves a 1 when it is combined with voxilaprevir. The color scale at the top shows distances in angstroms (from 10 Å to 30 Å), going from white to black. White means distances of 10 Å or less, suggesting a very close connection or direct interaction. Black means distances of 30 Å or more, indicating a large separation or no direct interaction between the amino acid and the membrane. Different shades of gray show distances between 10 Å and 30 Å, with darker shades meaning shorter distances.

**Ves a 1 alone with membrane:** In the simulation of Ves a 1 alone interacting with the membrane, several important parts of Ves a 1, including the crucial “lids” (T83–E93, T115–A135, and G251–G261), the key amino acids in the catalytic center (S170, D198, and H263), and the auxiliary site amino acids (F241, Y242, N244, and Q249), showed different levels of closeness to the membrane. The darker spots, especially those that are almost black, mean shorter distances, showing that these parts interacted more closely and consistently with the lipid membrane. For example, the dark patches suggest distances closer to 10Å. Importantly, the T83–E93 and T115–A135 lids, as well as the G251–G261 region, exhibited strong interactions (darker shades), which are likely very important for the enzyme to attach to the membrane and work properly. Areas that stayed white or very light-gray throughout the simulation indicated parts that remained 10 Å or more away from the membrane.

**Ves a 1 + Voxilaprevir with membrane:** When voxilaprevir was added, we saw clear changes in how Ves a 1 interacted with the membrane. While some interactions remained, changes in how strong or how long certain amino acids connected were shown by changes in color intensity. For example, if some regions became further away from the membrane, the corresponding areas in the heatmap became lighter (closer to white), meaning the shortest distance between the amino acid and the membrane increased towards 30 Å (e.g., from dark-gray ∼15 Å to light-gray ∼25 Å).

This suggests that the enzyme’s position shifted or its ability to bind to the membrane decreased because of the inhibitor. On the other hand, if interactions increased in other areas, they became darker, meaning the shortest distance decreased towards 10 Å, suggesting that voxilaprevir helped stabilize a shape that favors membrane binding. These color changes clearly show how voxilaprevir changed the way Ves a 1 relates to the lipid membrane. Figure 4B provides a visual look at the final state of the simulation at 1 µs, showing how Ves a 1 is positioned near the lipid membrane, both with and without voxilaprevir.

**Ves a 1 alone with membrane:** This image showed how Ves a 1 placed itself on the lipid membrane. The catalytic center (S170, D198, and H263) and auxiliary site (F241, Y242, N244, and Q249) were highlighted, along with the important lids (T83–E93, T115–A135, and G251–G261). Its position suggests how the enzyme might present its active site to those molecules it needs to act on, either within or on the membrane. The different colors (blue for stiff parts; red for flexible parts) gave us clues about how the protein moves. Flexible parts are likely involved in changing shape to bind molecules or release products.

**Ves a 1 + Voxilaprevir with membrane:** The presence of voxilaprevir (cyan color) clearly showed where it binds to Ves a 1. The way the Ves a 1-voxilaprevir combination sat on the membrane was different from Ves a 1 by itself. This visual comparison helped us understand how the inhibitor’s binding affected the enzyme’s overall shape and how it interacted with the membrane. If voxilaprevir binding made the enzyme stiffer or changed how the catalytic site was exposed to the membrane, it could explain how it stops the enzyme from working. The inhibitor’s location relative to the active site and the membrane is very important for understanding how well it works.

Overall, the results from Figure 4 showed that Ves a 1 interacts actively with the lipid membrane and that voxilaprevir affects this interaction. The distance analysis in Figure 4A, shown by the white-to-black color gradient representing distances from 10–30 Å, pointed out specific amino acids and regions involved in attaching to the membrane. The visualization in Figure 4B gave a clear picture of these interactions. The changes we saw in membrane interaction when voxilaprevir-bound suggest that the inhibitor not only targets the active site but also changes how the enzyme interacts with the membrane in a broader way, potentially impacting its overall function. Looking more closely at the specific amino acids involved in these altered membrane interactions could provide a deeper understanding of how voxilaprevir works.

In Figure 5, we present boxplots that show the catalytic area of Ves a 1 under different conditions. This area was measured from the center of three key regions: T83–E93, T115–A135, and G251–G261. This figure gave us exact information about how the catalytic area of Ves a 1 changes shape under different conditions, specifically when interacting with the lipid membrane, with or without the inhibitor voxilaprevir. The top graph shows Ves a 1 with a full membrane, while the bottom graph shows Ves a 1 with voxilaprevir in the presence of a membrane.

**Ves a 1 with membrane:** The boxplots for Ves a 1 with a membrane showed the catalytic area when the enzyme was interacting with a complete lipid membrane. We saw that the median catalytic area for Ves a 1 with the membrane was about $125\;{Å}^{2}$ across different runs. The full membrane tended to make the values cluster more tightly, suggesting a more fixed shape for the catalytic part.

**Ves a 1 + Voxilaprevir with membrane:** This section depicted the catalytic area of the complex when it was interacting with the lipid membrane. When we looked at the Ves a 1–Voxilaprevir complex, the catalytic area showed differences in its distribution. If voxilaprevir consistently led to a smaller or less varied catalytic area (for example, the median values for Ves a 1 + Voxilaprevir were slightly lower and less spread out than for Ves a 1 alone), this supports its role as an inhibitor that limits the shape changes needed for the enzyme to work.

Therefore, Figure 5 gives us clear, measurable information about how the catalytic area of Ves a 1 changes shape under different conditions when associated with a membrane. The catalytic area is defined by how the T83–E93, T115–A135, and G251–G261 regions are arranged, which are very important for the enzyme’s function and for its active site to be available. The comparison between Ves a 1 alone and the Ves a 1-Voxilaprevir combination, both in the membrane environment, highlights how important the lipid membrane is in shaping the enzyme’s form. Differences in the median and spread of the catalytic area distributions suggest that the membrane could cause specific shapes or limit how much the enzyme can change its shape.

Furthermore, the data revealed how voxilaprevir affected the catalytic area and how the membrane might change this effect. If voxilaprevir consistently led to a smaller or less varied catalytic area, it supported its role as an inhibitor that restricted the shape changes needed for the enzyme to work. The interplay between inhibitor binding and membrane interaction was critical; the membrane might have either enhanced or diminished the inhibitory effect by favoring certain conformational states. These findings are vital for understanding how the enzyme’s mechanism works in its native membrane environment and for designing more effective inhibitors.

Table 1 shows the calculated binding energies for Ves a 1 with voxilaprevir in the presence of a lipid membrane. These binding energies provide important numbers to understand how favorable these interactions are.

The binding energy for Ves a 1 with voxilaprevir in the presence of a membrane (Ves a 1 + Voxilaprevir (with membrane system)) was observed. The values for individual runs were −7.066 ± 0.51 kcal/mol (Run#1), −7.89 ± 0.45 kcal/mol (Run#2), and −7.59 ± 0.57 kcal/mol (Run#3), yielding an average $\Delta G_{\mathrm{Bind}}^{\mathrm{SIE}}$ of approximately −7.51 ± 0.51 kcal/mol. While data from systems without a membrane, such as Ves a 1 + ${\mathrm{DMPC}}_{a}$ and Ves a 1 + ${\mathrm{Voxilaprevir}}_{a}$, are available in the table for reference, direct quantitative comparisons of their binding energies with membrane-bound systems are inherently complex. The intricate environment of the lipid bilayer introduces multiple factors—including steric effects, altered conformational dynamics, and specific lipid-protein interactions—that are absent in simplified solution conditions.

These factors can significantly influence the enzyme’s binding site accessibility and configuration, making a direct ‘apples-to-apples’ comparison challenging. The observed binding strength in the membrane-bound system reflects the combined influence of the inhibitor and the physiological environment. Furthermore, the changes in the catalytic area seen in Figure 5 suggest that the membrane affects the overall shape of the enzyme, which could impact the exact structure needed for the inhibitor to bind best. Additionally, the hydrolysis process generates free fatty acids.

This remains a subject of future research, as our research did not include this information regarding the potential atomistic compromise of membrane integrity by free fatty acids following hydrolysis. This information could also be beneficial for understanding the behavior of Ves a 1 in the presence of free amino acids in the bilayer membrane environment. In conclusion, our research indicates a connection between the inhibitor, the enzyme, and their natural environment. This relationship is crucial to take into account when developing drugs for proteins that interact with membranes.

## 4. Conclusions

In summary, these findings offer a thorough and comprehensive understanding of how Ves a 1 moves and changes shape, how it interacts with the cell membrane, and the detailed ways in which voxilaprevir changes these interactions. Our work showed how important it is to consider the lipid membrane environment. This is crucial both for accurately understanding how this enzyme works and for wisely designing new drugs for Ves a 1 in the bilayer membrane environment. Based on these insights, voxilaprevir may be a promising candidate model for further development as a Ves a 1 inhibitor. 

## Figures and Tables

**Figure 1 toxins-17-00387-f001:**
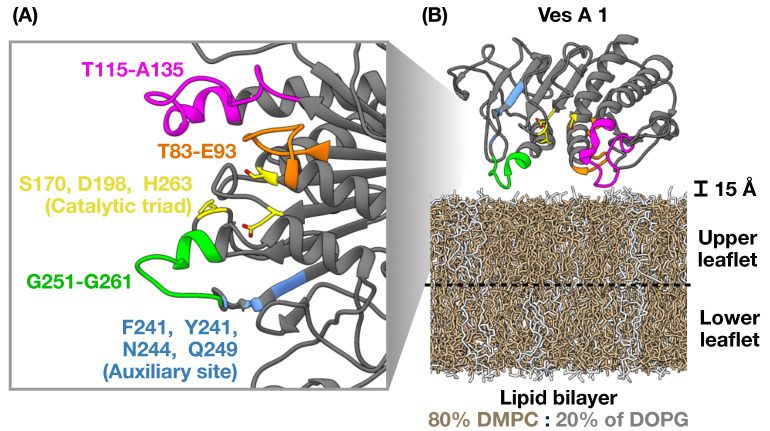
(**A**) The predicted structure of the Ves a 1 protein is shown with the catalytic triad (S170, D198, and H263) and auxiliary site (F241, Y242, N244, and Q249) in yellow and blue, respectively. The important lids T83-E93, T115-A135, and G251-G261 are in orange, magenta, and green, respectively. (**B**) The initial system configuration for an all-atom molecular dynamics simulation. The simple lipid bilayer contained 80% phosphatidylcholines (DMPC) and 20% phosphatidylglycerol (DOPG). Ves a 1 was initially placed about 15 Å above the upper leaflet of the bilayer.

**Figure 2 toxins-17-00387-f002:**
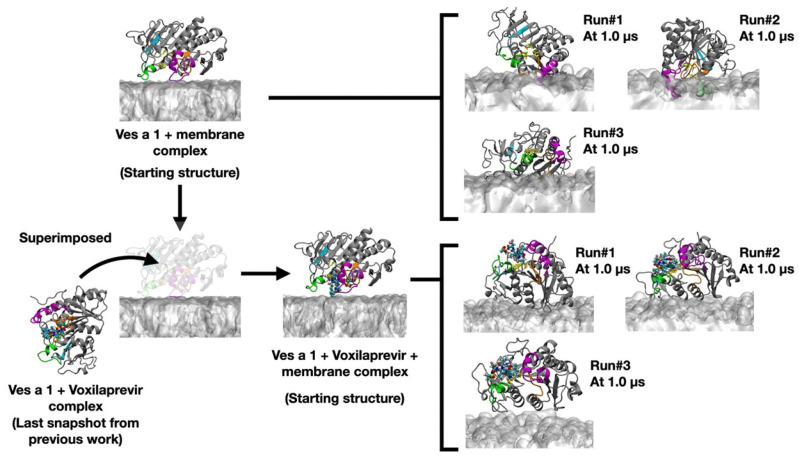
The process of preparing the ligand-bound Ves a 1 with a bilayer membrane. Ves a 1 + voxilaprevir was extracted from the final MD snapshot of the previous study of Ves a 1 with voxilaprevir in the solution [13]. The final MD snapshot was subsequently superimposed with respect to the modelled ligand-free Ves a 1 with bilayer membrane. The superimposed Ves a 1 + voxilaprevir + membrane complex was employed as the starting coordinate for the final three MD runs.

**Figure 3 toxins-17-00387-f003:**
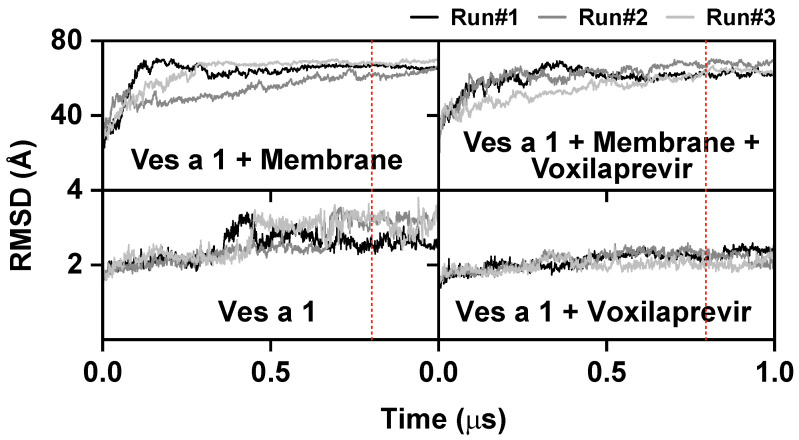
The plots of root-mean-square displacement (RMSD), where the black, gray, and light-gray lines represent Run#1, Run#2, and Run#3, respectively. The vertical red dotted line indicates the approximate point at 800 ns after which the systems reached a stable equilibrium.

**Figure 4 toxins-17-00387-f004:**
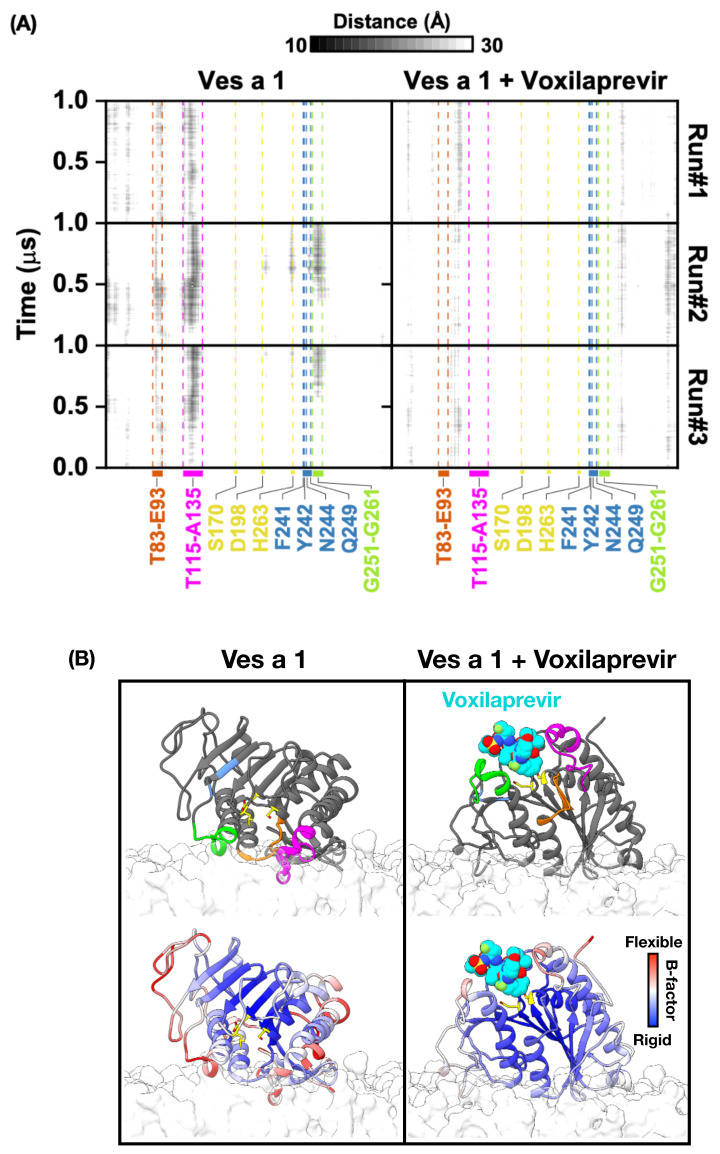
(**A**) Plot of the shortest distance between Ves a 1’s amino acid and phosphate group of the lipid on the lipid bilayer membrane. (**B**) The last trajectory at 1 µs shows the orientation of Ves a 1 interacting with the lipid bilayer membrane. Note that the catalytic triad (S170, D198, and H263) and auxiliary site (F241, Y242, N244, and Q249) are in yellow and blue, respectively. The important lids: T83–E93, T115–A135, and G251–G261 are in orange, magenta, and green, respectively. Voxilaprevir is coloured in cyan.

**Figure 5 toxins-17-00387-f005:**
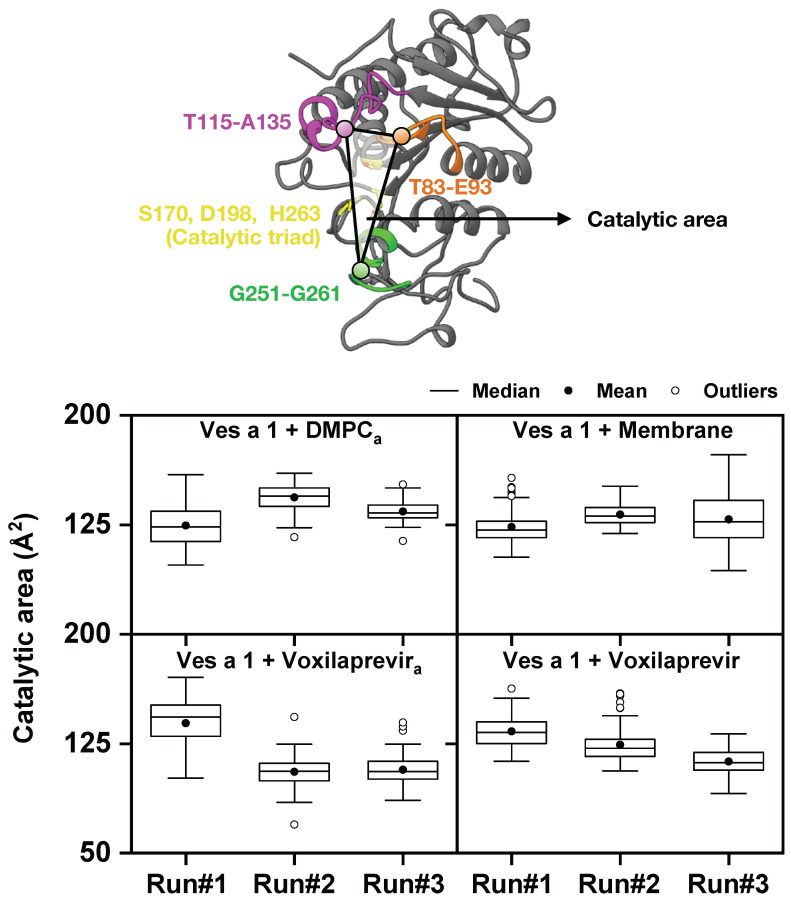
Boxplots illustrating the catalytic area of Ves a 1 under different conditions, calculated from the center of mass of three key regions: T83–E93, T115–A135, and G251–G261. Both boxplots contain the results from the membrane-free Ves a 1 and membrane-bound Ves a 1. The subscript *a* denotes systems without a membrane from a previous study [13]. The bottom right (Ves a 1 + Membrane) represents the Ves a 1 in the bilayer membrane condition (80% DMPC + 20% DOPG). The top panel compares Ves a 1 with ${\mathrm{DMPC}}_{a}$ and Ves a 1 with membrane, while the bottom panel compares Ves a 1 with ${\mathrm{voxilaprevir}}_{a}$ (membrane-free) and Ves a 1 with voxilaprevir in the presence of a membrane.

**Table 1 toxins-17-00387-t001:** Energy components (kcal/mol) calculated using the solvated interaction energy (SIE) method.

Ligands	Δ $E_{\mathrm{vdW}}$	Δ $E_{\mathrm{Ele}}$	Δ $G_{\mathrm{RF}}$	γΔSA	$\Delta G_{\mathrm{Bind}}^{\mathrm{SIE}}$
Ves a 1 + Voxilaprevir					
Run#1	−37.61 ± 4.89	−1.16 ± 1.90	5.97 ± 2.09	−7.00 ± 0.86	−7.066 ± 0.51
Run#2	−44.48 ± 3.92	−1.13 ± 2.41	6.49 ± 2.11	−8.65 ± 0.65	−7.89 ± 0.45
Run#3	−43.96 ± 4.84	−1.40 ± 2.67	9.43 ± 3.00	−8.89 ± 1.08	−7.59 ± 0.57
Ves a 1 + ${\mathrm{DMPC}}_{a}$					
Run#1	−56.20 ± 3.76	−14.40 ± 2.78	19.37 ± 2.36	−11.77 ± 0.50	−9.49 ± 0.43
Run#2	−41.97 ± 6.09	−3.80 ± 3.67	9.24 ± 2.86	−9.80 ± 1.44	−7.74 ± 0.64
Run#3	−53.36 ± 3.98	−19.14 ± 4.78	22.13 ± 3.91	−11.21 ± 0.70	−9.34 ± 0.51
Ves a 1 + ${\mathrm{Voxilaprevir}}_{a}$					
Run#1	−58.80 ± 3.60	−84.00 ± 6.63	84.88 ± 4.98	−11.45 ± 0.32	−10.16 ± 0.43
Run#2	−57.96 ± 4.59	−72.34 ± 11.32	75.11 ± 8.70	−10.91 ± 0.63	−9.81 ± 0.74
Run#3	−60.10 ± 3.03	−83.61 ± 9.49	84.43 ± 8.21	−11.52 ± 0.40	−10.31 ± 0.40

The *a* superscript denotes systems without a membrane from a previous study [13].

## Data Availability

The original contributions presented in this study are included in the article/Appendix A. Further inquiries can be directed to the corresponding author(s).

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
