# Peer review of "Atomistic-Level Structural Insight into Vespa Venom (Ves a 1) and Lipid Membrane Through the View of Molecular Dynamics Simulation"

_toxins, 2025, doi:10.3390/toxins17080387_

Round 1

Reviewer 1 Report

Comments and Suggestions for Authors

The manuscript describes atomistic-level binding insight of Vespa venom (Ves a 1) to lipid membrane through the view of molecular dynamics simulation. The topic is relevant to the aim and scope of the Toxins. The manuscript is well written and easy to follow. However, this manuscript meets the standard for acceptance after addressing the below comments:

  1. The simulated bilayer consists of 80% DMPC and 20% DOPG. How does this composition reflect the actual lipid composition of mammalian or insect membranes? Can the authors justify their choice, or comment on how results may vary with a more biologically relevant bilayer (e.g., with cholesterol or sphingolipids)? In other words, why should 80% DMPC and 20% DOPG be chosen as components of the bilayer model? Generally, PC is abundant in membrane. Then, what is the role of PG? How about PE, PS, or PA? Why are these excluded?

  1. The effect of voxilaprevir alone (without membrane) on Ves a 1 conformation is less discussed. Could the authors expand the discussion of how voxilaprevir influences Ves a 1 in solution, and whether membrane presence amplifies or masks inhibitor effects?

  1. Equation (1) uses empirical constants (α = 0.105, γ = 0.013, C = -2.89). Please briefly justify or cite the origin of these constants, particularly for readers unfamiliar with the SIE method..

  1. A ∆G_bind of −7.5 kcal/mol suggests moderate affinity. Can the authors contextualize this in relation to typical inhibitor strengths (e.g., nanomolar vs. micromolar binding)?

  1. As far as I understand, free fatty-acids are generated by hydrolysis. Then, is there no further interaction caused by these fatty-acids to somewhere?

Author Response

Reviewer 1:

Thanks for carefully reading our manuscript and the constructive comments. Please find the responses of the comments follows:

Comment 1: The simulated bilayer consists of 80% DMPC and 20% DOPG. How does this composition reflect the actual lipid composition of mammalian or insect membranes? Can the authors justify their choice, or comment on how results may vary with a more biologically relevant bilayer (e.g., with cholesterol or sphingolipids)? In other words, why should 80% DMPC and 20% DOPG be chosen as components of the bilayer model? Generally, PC is abundant in membrane. Then, what is the role of PG? How about PE, PS, or PA? Why are these excluded?

Response:

I appreciate your recommendation. To address this, the bilayer membrane utilized in the simulation should be consistent with the organism; however, numerous human membrane models were implemented in numerous investigations. We acknowledged that we are not the experts in determining the full impact of these PG, PE, PS, or PA on each bilayer membrane. Additionally, the results could be inconsistent across different models due to the component's potential to alter membrane fluidity, however the previous review suggest that the simplified bilayer model is still good enough to investigate the biological system via molecular dynamics, according to the review in 2019

  • Marrink SJ, Corradi V, Souza PC, Ingolfsson HI, Tieleman DP, Sansom MS. Computational modeling of realistic cell membranes. Chemical reviews. 2019 Jan 9;119(9):6184-226.

Instead of engaging in this debate, we implemented the simple bilayer membrane model (80% DMPC and 20% DOPG) to represent this model, which is in accordance with the previous investigation of the M37 lipase simulation.

  • Willems N, Lelimousin M, Koldsø H, Sansom MS. Interfacial activation of M37 lipase: A multi-scale simulation study. Biochimica et Biophysica Acta (BBA)-Biomembranes. 2017 Mar 1;1859(3):340-9.

We have addressed this point in the revised manuscript in the Parameterisation section, to inform the reader in case of question, as followings:

“To address this, the bilayer membrane utilized in the simulation should be consistent with the organism. Numerous human membrane models were employed, and the component’s capacity to modify membrane fluidity may result in inconsistent results across various models. However, to avoid these issues, the simplified bilayer model is still sufficient for the molecular dynamics investigation of the biological system [19].

To represent this model, we implemented the simple bilayer membrane model (80% DMPC and 20% DOPG). The simple lipid bilayer model was built based on the mixture between 80% of phosphatidylcholines (DMPC) and 20% of phosphatidylglycerol (DOPG) similar to previous studies [20,25].”

Comment 2: The effect of voxilaprevir alone (without membrane) on Ves a 1 conformation is less discussed. Could the authors expand the discussion of how voxilaprevir influences Ves a 1 in solution, and whether membrane presence amplifies or masks inhibitor effects?

Response:

Thanks for suggestion. We have changed the scope of the study in the last paragraph of the introduction. As this study focused on voxilaprevir-Ves a 1 in the membrane environment, we did not discuss membrance-free Ves a 1. The membrane-free Ves a 1 was reported before in the previous study

  • Pattaranggoon NC, Daduang S, Rungrotmongkol T, Teajaroen W, Tipmanee V, Hannongbua S. Computational model for lipid binding regions in phospholipase (Ves a 1) from Vespa venom. Scientific Reports. 2023 Jun 30;13(1):10652.

To inform the reader for this issue, we have added the sentence “In this study, we focused on the Ves a 1 under the bilayer membrane condition. The membrane-free Ves a 1 was investigated before in the previous study.” in the first paragraph of the result and discussion.

Also, we have added the importance of membrane inclusion to the system by adding the sentence “Despite these findings, the detailed interaction and behaviors of Ves a 1 under lipid membrane bilayer remain insufficiently understood. The membrane environment inclusion is a key factor for fully understanding of membrane-embedded protein both in ligand-free and ligand-bound states [18–20].” in the introduction, to justify why we need to simulate the Ves a 1 in the membrane. In addition, this sentence can respond “whether membrane presence amplifies or masks inhibitor effects”, in the introduction section,

Thanks very much again. This improves the clarity and justification of what we do to the reader. We appreciated this very much.

Comment 3: Equation (1) uses empirical constants (α = 0.105, γ = 0.013, C = -2.89). Please briefly justify or cite the origin of these constants, particularly for readers unfamiliar with the SIE method. 

Response:

Thanks for suggestion. We have added the references for these values as followed:

  • Thirunavukkarasu, M.K.; Suriya, U.; Rungrotmongkol, T.; Karuppasamy, R. In silico screening of available drugs targeting non-small cell lung cancer targets: A drug repurposing approach. Pharmaceutics 2021, 14, 59.
  • Somboon, T.; Mahalapbutr, P.; Sanachai, K.; Maitarad, P.; Lee, V.S.; Hannongbua, S.; Rungrotmongkol, T. Computational study on peptidomimetic inhibitors against SARS-CoV-2 main protease. Journal of molecular liquids 2021, 322, 114999.

We rewrote the sentence citing these references, “where ∆EvdW is van der Waals, γ∆SA is cavity, ∆EEle is electrostatic, and ∆GRF is reaction field. ∆EvdW and ∆EEle are denoted as intermolecular energies in the bound state. The coefficients used in every calculation are α, γ, and C as 0.105, 0.013, and -2.89, respectively, similar to the previous studies [26,27].” in the manuscript.

Comment 4: A ∆Gbind of −7.5 kcal/mol suggests moderate affinity. Can the authors contextualize this in relation to typical inhibitor strengths (e.g., nanomolar vs. micromolar binding)

Response:

Thanks for suggestion. We have removed this sentence as the word “moderate’ and to quantify this is very subjective and can confuse the reader.

Comment 5: As far as I understand, free fatty-acids are generated by hydrolysis. Then, is there no further interaction caused by these fatty-acids to somewhere?

Response:

Thanks for suggestion. We have focused on the binding the voxilaprevir to Ves a 1 in the membrane environment. Truly, we did not study the state of Ves a1 after free fatty acids are generated. We do not discuss and include this information in this manuscript. This reminds us that it could be the next point to study in detail, as the experiment reported the free fatty acids after hydrolysis could compromise membrane integrity.

To respond this question, not only the reviewer but also the reader, we have added the sentence

“Additionally, the hydrolysis process generates free fatty acids. This remains a subject of future research, as our research did include this information regarding the potential atomistic compromise of membrane integrity by free fatty acids following hydrolysis. This information could also be beneficial in understanding the behavior of Ves a 1 in the presence of free amino acids in the bilayer membrane environment. In conclusion, our research indicates a connection between the inhibitor, the enzyme, and their natural environment. This relationship is crucial to take into account when developing drugs for proteins that interact with membranes” in the last part of the manuscript to state the limit of this study and the next possibility for the further study.

Reviewer 2 Report

Comments and Suggestions for Authors

In the manuscript entitled 'Atomistic-level Binding Insight of Vespa Venom (Ves a 1) to Lipid Membrane Through The View of Molecular Dynamics Simulation', the authors employ full-atomistic MD simulations (with the Amber package) to investigate the behavior of Vespapase in and outside a PC:PG (8:2) membrane environment, and in the presence of Voxilaprevir. The manuscript is well-written, although I did observe slight over-advertising of results in the introduction. For instance, the authors state that 'the behavior of Ves a 1 both in the presence and absence of a lipid bilayer' or 'The aim of this study is to provide mechanistic insights into how voxilaprevir influences Ves a 1 activity'. The authors do not appear to perform either of these tasks; rather, they observe differences in structural conformations and calculate total binding free energy. While these are still important findings, the final paragraph of the Introduction should be toned down a little to accurately reflect the scope of the study.

I did not identify any major methodological issues; three separate runs of each system are carried out. However, I am a little concerned regarding Ves a 1 binding to the membrane. The incorporation of such large structures into a membrane is quite challenging in MD, as the protein often ignores the membrane, especially given that the authors state there is a 15 Å distance between Ves a 1 and the membrane. I assume that incorporation was achieved by implementing NVT conditions. Was the volume of the system set in such a way that the membrane was "loose" enough to facilitate incorporation? Or were other steps implemented to ensure effective incorporation? I am also concerned that, according to Figure 3, Ves a 1 performed a complete 180-degree flip when Voxilaprevir was in the system. I did not find information on how this specific system was designed – specifically, was Ves a 1 + Voxilaprevir (already bound) placed above the membrane, or was Voxilaprevir added to a system where Ves a 1 was already bound to the membrane? Alternatively, were all three objects added to the system separately? The authors need to describe this part in more detail.

My minor remarks: 
- I found only scarse information about the Voxilaprevir in the introduction. It was stated that it is a proposed protease inhibitor (it is often called 'drug' in later manuscript). Why this drug? What is the reason for selecting it?

- Fig 4. Bottom right panel description might be altered to include information about the membrane. 

- The authors should use an open system to deposit either their trajectories or initial and final system states in repositories such as Zenodo. The statement 'All data have been provided in this manuscript.' is not sufficient for an MD simulation study.

- Issues with citation: Spangfort, M.D.; et al. Structure and biology of stinging insect venom allergens. International archives of allergy and immunology 2000, 123, 99–106. According to PubMed (https://pubmed.ncbi.nlm.nih.gov/11060481), the correct authors are King and Spangfort.

- The authors should provide a confidence level of prediction if using AlphaFold. At least one value such as ipTM, PAE, or pLDDT should be provided.

Author Response

Reviewer 2:

In the manuscript entitled 'Atomistic-level Binding Insight of Vespa Venom (Ves a 1) to Lipid Membrane Through The View of Molecular Dynamics Simulation', the authors employ full-atomistic MD simulations (with the Amber package) to investigate the behavior of Vespapase in and outside a PC:PG (8:2) membrane environment, and in the presence of Voxilaprevir. The manuscript is well-written, although I did observe slight over-advertising of results in the introduction.

Response: Thanks very much. We appreciated this.

Comment 1: For instance, the authors state that 'the behavior of Ves a 1 both in the presence and absence of a lipid bilayer' or 'The aim of this study is to provide mechanistic insights into how voxilaprevir influences Ves a 1 activity'. The authors do not appear to perform either of these tasks; rather, they observe differences in structural conformations and calculate total binding free energy. While these are still important findings, the final paragraph of the Introduction should be toned down a little to accurately reflect the scope of the study.

Response: Thanks very much for these helpful comments.

We clearly understand this point and overclaim it. Therefore, we agree to tone down the scope of the study as following:

  1. We have renamed the article into “Atomistic-level Structural Insight of Vespa Venom (Ves a 1) to Lipid Membrane Through The view of Molecular Dynamics Simulation”.
  2. We have rewritten the sentence 'the behavior of Ves a 1 both in the presence and absence of a lipid bilayer' to ‘the structural information of Ves a 1 in the presence of lipid bilayer’
  3. We have rewritten the sentence 'The aim of this study is to provide mechanistic insights into how voxilaprevir influences Ves a 1 activity' to ‘The aim of this study is also to provide structural insights of how voxilaprevir bind Ves a 1 and how this binding may influence its action on the lipid substrate, particularly in membrane-associated environments, thereby supporting the potential development of Ves a 1-targeted inhibitors.

Comment 2: I did not identify any major methodological issues; three separate runs of each system are carried out. However, I am a little concerned regarding Ves a 1 binding to the membrane. The incorporation of such large structures into a membrane is quite challenging in MD, as the protein often ignores the membrane, especially given that the authors state there is a 15 Å distance between Ves a 1 and the membrane. I assume that incorporation was achieved by implementing NVT conditions. Was the volume of the system set in such a way that the membrane was "loose" enough to facilitate incorporation? Or were other steps implemented to ensure effective incorporation?

Response:

We did not bury the protein in the bilayer membrane. We just put the protein above the membrane. The volume of the system was equilibrated in the NVT, and the incorporation was achieved by implementing NPT conditions. We assume the membrane was loose enough to facilitate incorporation as the protein just attached the membrane surface not buried in the bilayer. In addition, we use the simple membrane model which was reported and used in the previous studies, so we assume that this model is good enough to represent these simulated systems.

Comment 3: I am also concerned that, according to Figure 3, Ves a 1 performed a complete 180-degree flip when Voxilaprevir was in the system. I did not find information on how this specific system was designed – specifically, was Ves a 1 + Voxilaprevir (already bound) placed above the membrane, or was Voxilaprevir added to a system where Ves a 1 was already bound to the membrane? Alternatively, were all three objects added to the system separately? The authors need to describe this part in more detail.

Response:

The structure of Ves a 1 + Voxilaprevir (already bound) was modelled using the superimposition to the Ves a 1 (ligand-free) placed above the membrane. The last MD snapshot of Ves a 1 + Voxilaprevir from previous study was taken to be superimposed to generate a Ves a 1 + Voxilaprevir + membrane complex, a starting coordinate for the three MD runs. The figure illustrated how to model these systems was included in the revised manuscript, along with the sentence in the computational method section as “The structure of Ves a 1 + voxilaprevir was modeled by superimposing it on the ligand-free Ves a 1 placed above the membrane. The final MD snapshot of Ves a 1 + voxilaprevir from the previous study was selected for superimposition in order to produce a Ves a 1 +voxilaprevir + membrane complex, which served as the initial coordinate for the three MD runs. The system preparation process, both ligand-free and ligand-bound Ves a 1 structures, was illustrated in Figure 2.

The schematic figure of the process was illustrated in Figure 2, along with the caption in the revised manuscript.

Figure 2. The process of preparing the ligand-bound Ves a 1 with a bilayer membrane. The Ves a 1 + voxilaprevir was extracted from the final MD snapshot of the previous study of Ves a 1 with voxilaprevir in the solution [12]. The final MD snapshot was subsequently superimposed with respect to the modelled ligand-free Ves a 1 with bilayer membrane. The superimposed Ves a 1 + voxilaprevir + membrane complex was employed as the starting coordinate for the final all three MD runs.

Minor remark 1: I found only scarse information about the Voxilaprevir in the introduction. It was stated that it is a proposed protease inhibitor (it is often called 'drug' in later manuscript). Why this drug? What is the reason for selecting it?

Response:

Thanks for suggestion. We are sorry to not include this important information of Voxilaprevir used in this study. In 2023, our previous investigation of Ves a 1 in the solution revealed that Voxilaprevir were one of the top candidates for binding Ves a 1 among 2,056 FDA-approved drugs and a ligand control (1,2-Dimyristoyl-sn-glycero-3-phosphocholine or DMPC) at both the catalytic and auxiliary sites. The top five drug candidates with catalytic site complexes were chosen: DMPC (PubChem CID 5459377), doxycycline (ZINC16052277), atovaquone (ZINC100017856), Ubrogepant (ZINC95598454), and voxilaprevir (e-Drug3D ID D1847). We determined to use voxilaprevir as the repurposed drug, which could serve as an inhibitor.

We have added the sentence “The repurposed drug became of interest in the drug discovery including an enzyme inhibitor. We also adopted this concept in the study of Ves a 1 inhibitor exploration. Our previous investigation of Ves a 1 in the solution with 2,056 FDA-approved drugs and a ligand control (1,2-Dimyristoyl-sn-glycero-3-phosphocholine or DMPC) revealed that voxilaprevir were one of the top candidates for binding Ves a 1 among at both the catalytic and auxiliary sites. We decided to use a voxilaprevir, protease inhibitor, as a repurposed drug candidate.” in the introduction part to respond this comment.

Minor remark 2: Fig 4. Bottom right panel description might be altered to include information about the membrane. 

Response:  We have corrected into “The boxplots illustrating the catalytic area of Ves a 1 under different conditions, calculated from the center of mass of three key regions: T83-E93, T115-A135, and G251-G261. Both boxplots contained the results from the membrane-free Ves a 1 and membrane-bound Ves a 1. The subscript a denoted systems without a membrane from a previous study. The bottom right (Ves a 1 + Membrane) represented the Ves a 1 in the bilayer membrane condition (80% DMPC + 20% DOPG).”.

Minor remark 3: The authors should use an open system to deposit either their trajectories or initial and final system states in repositories such as Zenodo. The statement 'All data have been provided in this manuscript.' is not sufficient for an MD simulation study.

Response: We have included the PDB files of the initial states of the Ves a 1 and ligand-Ves a 1. Also, we have provided the PDB files of the last MD snapshots of both states in all three runs. These files were available in the supplementary materials. The sentence “The Ves a 1 and voxilaprevir-Ves a 1 structure files included in this study were available in PDB format. The initial structure, last MD snapshots in all three runs of Ves a 1, and voxilaprevir-Ves a 1 were provided in the PDB files.”  was added.

Minor remark 4: Issues with citation: Spangfort, M.D.; et al. Structure and biology of stinging insect venom allergens. International archives of allergy and immunology 2000, 123, 99–106. According to PubMed (https://pubmed.ncbi.nlm.nih.gov/11060481), the correct authors are King and Spangfort.

Response: We have corrected this reference as

  • King, T.P.; Spangfort, M.D. Structure and biology of stinging insect venom allergens. International archives of allergy and immunology 2000, 123, 99–106.

Minor remark 5: The authors should provide a confidence level of prediction if using AlphaFold. At least one value such as ipTM, PAE, or pLDDT should be provided.

Response: We have responded this comment by adding this sentence in the paramerisation section as

“The amino acid sequences of Ves a 1 was taken from Uniprot database (UniProtKB code: P0DMB4) and then submitted to AlphaFold 2 for structure prediction. The 3D structure of Ves a 1 was predicted based on the UniProt entry (ID: P0DMB4), with the corresponding AlphaFold model ID: AF-P0DMB4-F1-v4. The per-residue confidence score (pLDDT) indicated very high confidence (pLDDT > 90). The prediction result was provided in the supplementary materials, Figure S1.”

Round 2

Reviewer 1 Report

Comments and Suggestions for Authors

All issues have been addressed.

Reviewer 2 Report

Comments and Suggestions for Authors

Authors responded to all my remarks and addressed them in details. I recommend accepting the MS.